# Fabrication of Co-Based Cladding Layer by Microbeam Plasma and Its Corrosion Mechanism to Molten Salt

**DOI:** 10.3390/ma17174249

**Published:** 2024-08-28

**Authors:** Kaiqi Sun, Yufeng Zhang, Yingfan Wang, Fuxing Ye

**Affiliations:** 1School of Materials Science and Engineering, Tianjin University, Tianjin Key Laboratory of Advanced Joining Technology, Tianjin 300072, China; sunkq0413@tju.edu.cn (K.S.); 2021208079@tju.edu.cn (Y.Z.); 3018208222@tju.edu.cn (Y.W.); 2Key Lab of Advanced Ceramics and Machining Technology of Ministry of Education, Tianjin 300072, China

**Keywords:** Co-based cladding layer, NiCr-Cr_3_C_2_, molten salts Na_2_SO_4_ and NaCl corrosion, friction

## Abstract

Corrosion of the molten salts Na_2_SO_4_ and NaCl has become one of the major factors in the failure of steel components in boilers and engines. In this study, CoNiCrAlY cobalt-based cladding layers with different NiCr-Cr_3_C_2_ ratios were prepared by microbeam plasma cladding technology. The influence of the NiCr-Cr_3_C_2_ content on the microstructure, mechanical properties, and molten salt corrosion resistance of CoNiCrAlY was investigated. The CoNiCrAlY with a 25 wt.% NiCr-Cr_3_C_2_ (NC25) cladding layer possessed the highest microhardness (348.2 HV_0.3_) and the smallest coefficient of friction (0.4751), exhibiting great overall mechanical properties. The generation of protective oxides Cr_2_O_3_, Al_2_O_3_, and spinel phase (Ni,Co)Cr_2_O_4_ is promoted by the addition of 25 wt.% NiCr-Cr_3_C_2_, which significantly reduces the corrosion of the cladding layer, and this effect is much more obvious at 950 °C than that at 750 °C. Furthermore, its corrosion mechanism was clarified. From the findings emerge a viable solution for the design and development of new high-temperature corrosion-resistant coatings.

## 1. Introduction

Inconel 625 is a typical nickel-based high-temperature alloy that is widely used in high-temperature environments such as industrial gas turbines. However, with the rapid development of modern industry, components are subjected to increasingly demanding service conditions. At a high temperature of about 750 °C, steel components can wear severely because of softening and oxidation, and are often corroded in reaction with combustion gas and deposited salts, which can lead to material failure and significant economic losses [1,2,3]. Among various approaches, the most cost-effective and efficient approach to solve this problem is to coat them with wear- and corrosion-resistant coatings. The coating provides protection against wear and corrosion, extending the service life of the material, and the substrate provides the necessary mechanical strength [4,5,6].

The material and structural design of the cladding layer is important for protecting the substrate material. CoNiCrAlY coatings have recently received considerable attention because of their excellent properties such as oxidation and corrosion resistance, thanks to the reasonable elemental composition [7,8,9]. The high oxidation and corrosion resistance of NiCr-Cr_3_C_2_, along with a similar thermal expansion coefficient to the substrate Inconel 625, are extensively used to fabricate wear- and corrosion-resistant coatings on steel components used in the hot sections of various boilers and engines [10,11,12,13]. In order to improve the particle erosion resistance of CoNiCrAlY cladding layer, different contents of NiCr-Cr_3_C_2_ were added in this study, and the influence law of the additive amount on the wear and molten salt corrosion resistance of the cladding layer was systematically investigated. Critically, the functioning of the metal components is significantly hindered by the attacks of molten salts, such as Na_2_SO_4_ and NaCl, caused by gas burning, leading to corrosion and failure of the components [14,15]. Xia et al. found that Na_2_SO_4_ is a dominant force in corrosion, but the addition of even small amounts of NaCl in Na_2_SO_4_ is much more harmful than only either NaCl or Na_2_SO_4_ [16]. Wang et al. found that chlorine produced by the reaction of NaCl molten salt and O_2_ is one of the main thermal corrosion factors for components [17]. Thermal corrosion is primarily categorized into low-temperature (650~800 °C) and high-temperature (850~950 °C) thermal corrosion, and the peak corrosion rates of NiCrAl alloys are about 700 °C and 900 °C, respectively [18,19]. However, the higher amount of Co contained in the CoNiCrAlY powder used in this paper would theoretically lead to a higher corrosion resistance of the cladding layer, and, therefore, the corrosion experiments were set at the slightly higher temperatures of 750 °C and 950 °C.

The present study uses microbeam plasma cladding technology to prepare cobalt-based cladding layers with good mechanical properties and molten salt corrosion resistance. The optimal composition of the cladding layer was determined, high-temperature molten salt corrosion experiments were conducted, and the corrosion mechanism was clarified. The findings can aid in determining a viable solution for the design and development of high-temperature corrosion-resistant coatings that will extend the service life of equipment and improve production efficiency.

## 2. Materials and Methods

The CoNiCrAlY powder used in this experiment has particle sizes of 170~325 mesh, predominantly consisting of γ/γ′ and β-NiAl phases, with the chemical composition presented in Table 1. The NiCr-Cr_3_C_2_ powder has particle sizes of 325~500 mesh, comprising 25 wt.% NiCr and 75 wt.% Cr_3_C_2_ phases.

Microbeam plasma cladding technology was utilized to prepared the cobalt-based cladding layers on Inconel 625 substrates. The cladding layer without additives is denoted as CoNiCrAlY, while layers with 5, 10, 15, 20, 25, and 30 wt.% of NiCr-Cr_3_C_2_ are denoted as NC5, NC10, NC15, NC20, NC25, and NC30, respectively. The powder compositions and proportions are outlined in Table 2.

The mechanical alloying process via ball milling was conducted with a rotation rate of 500 RPM, a ball-to-material ratio of 10:1, and alternating positive and negative rotations for two hours. After ball milling, the powder was compacted and flattened by placing it into a mold with PVA, producing a rectangular powder strip measuring 50 mm × 4 mm × 2 mm. Subsequently, the strip was dried at 120 °C for two hours in an oven for further utilization. The substrates used were Inconel 625 plates, and the surface was polished with sandpaper to remove the oxide film. The grease was washed off, and the surface was cleaned with alcohol and oven-dried at 80 °C for 1 h.

The experiment utilized a transTIG 800 welding power supply manufactured by Fronius Company (Wels, Austria), and the built-in Plasma Module 10 was employed as a microbeam plasma arc generator. The transTIG is a fully digitized TIG DC power source with a welding current range of 0.5 A to 80 A [20]. The plasma and shielding gas used in the experiment was 99.99% pure argon gas. To facilitate welding, the prepared powder strip was positioned on the central axis of the substrate surface. After experimentation and optimization, the welding process’s parameters were set as shown in Table 3.

This work employed a field emission scanning electron microscope (SEM, JSM-7800F, JEOL Ltd., Tokyo, Japan) with X-ray energy dispersive spectroscopy (EDS) to observe the microstructure, element distribution, and grain size. In addition, X-ray diffraction (XRD, D8 Advance, Bruker AXS, Karlsruhe, Germany) using Cu-K_α_ radiation was applied to detect the phase structure of cladding layers, as well as whether a phase change occurs after wear and molten salt corrosion. Rietveld whole-spectrum fitting analysis, a crystal structure refinement method, was performed using EXPGUI GSAS 1.00 software to quantitatively analyze the phase composition of the cladding layer.

The microhardness of the cladding layer was measured using a Vickers hardness tester (HV-1000A, Huayin Test Instrument Co. Ltd., Laizhou, China). The friction and wear experiments on the cladding layer were conducted at room temperature using the UMT-2 reciprocal sliding friction and wear testing apparatus provided by The Bruker Company (Ettlingen, Germany). The tribological test parameters are shown in Table 4. A white light interferometer 3D surface profiler was used to measure the wear scars’ depth and width to calculate the cladding layer’s wear rate. The wear rate *R* is calculated using the following formula:(1)$$R=\frac{V}{F\times L}$$
where the *V* is the wear volume (mm^3^), the *F* is the load (N), and the *L* is the wear distance (m).

High-temperature molten salt corrosion tests were conducted in this experiment. Multiple rectangular samples, measuring 20 mm × 10 mm × 2 mm, were cut from the cladding layer, using wire-cutting technology, polished, cleaned, and dried to obtain more accurate corrosion data. To simulate the service conditions of the cladding layer, a corrosive mixture of 75 wt.% Na_2_SO_4_ + 25 wt.% NaCl was selected. The salt mixture was applied on the sample surface at 3 mg/cm^2^, and molten salt corrosion experiments were conducted at 750 °C and 950 °C. The samples were taken out every 5 h to calculate the cladding layer’s corrosion weight loss per unit area, thereby evaluating the molten salt corrosion resistance. After 50 h of continuous molten salt corrosion experiments, the corrosion products were analyzed to investigate the corrosion resistance mechanism of the cladding layer.

## 3. Results and Discussion

### 3.1. Microstructures

Figure 1 presents the X-ray diffraction (XRD) patterns of NC5~NC30 cladding layers. The main phases in the cladding layer include the γ/γ′ phases, β-NiAl, and the following three types of chromium carbide: Cr_3_C_2_, Cr_7_C_3_, and Cr_23_C_6_. After the Rietveld fitting and quantitative analysis, the phase content proportions in each cladding layer were calculated, as shown in Table 5. It is evident that the Cr_3_C_2_ and β-NiAl contents in the NC25 cladding layer are the highest, indicating a higher bonding density and stronger resistance to deformation. Consequently, it is inferred that NC25 has the best overall mechanical properties.

Figure 2 presents the microstructure and elemental distribution of the NC25 cladding layer. The spherical gray phase is primarily composed of Fe, Co, and Ni, which share the same crystal structure at elevated temperatures. According to the XRD analysis results in Figure 1, a γ/γ′ solid solution formed in the Ni. The black particulate matter contained in the spherical gray phase is characterized by dispersed aluminum enrichment. During oxidation or corrosion, the incoming oxygen is consumed by these aluminum-rich areas, resulting in the formation of Al_2_O_3_, which provides protective properties. Combining the XRD analysis results in Figure 1, the grain boundary is identified as the β phase, which is primarily a solid solution of Ni and Al.

The magnified microstructure of the intergranular regions is exhibited in Figure 2b, and the EDS spectrum of the white phase is shown in Figure 2c. Combined with the XRD analysis results in Figure 1, the white phase is mainly composed of chromium carbide particles. Moreover, it is assumed that in addition to the added Cr_3_C_2_, new phases Cr_7_C_3_ and Cr_23_C_6_ are generated within the chromium carbide particles. The wear resistance is enhanced by these high-hardness chromium carbide particles. Furthermore, their intimate association with the β phase matrix contributes to the overall performance of the cladding layer, particularly in improving corrosion resistance.

Figure 3 illustrates the microstructures of the NC5~NC30 cladding layers. The phase compositions of each sample were similar. Combining the above phase composition analyses, it can be seen that with the increase in NiCr-Cr_3_C_2_ contents, the width of the β phase at the grain boundaries increased, and the size of the gray grains increased and then decreased. The change in the NiCr-Cr_3_C_2_ contents affected the distribution of chemical elements within the molten pool, thereby influencing grain growth and morphology.

### 3.2. Hardness and Tribological Properties

The microhardness curve in Figure 4 represents a top-down analysis of the CoNiCrAlY and NC5~NC30 cladding layer sections, revealing that the hardness distribution of the cladding layers were relatively uniform. The average hardness of NC25 was the highest, reaching 348.2 HV_0.3_, an increase of 17.8% compared to the substrate, while the hardness of the NC30 cladding layer significantly decreased. The added NiCr-Cr_3_C_2_ undergoes decarburization during the cladding process. As the NiCr-Cr_3_C_2_ content increases, the carbon content in the alloy rises, the decarburization phenomenon is suppressed, and the Cr_3_C_2_ content gradually increases, improving the cladding layer’s hardness. However, an excess of NiCr-Cr_3_C_2_ will promote the decomposition of the Cr_3_C_2_ phase, reducing the hardness of the cladding layer.

The average coefficient of friction (COF) and wear rate for each sample are illustrated in Figure 5a, highlighting the similarities observed between these two parameters. The 2D images of wear scar morphology are shown in Figure 5b. It can be seen that the wear resistance of NC5~NC20 is inferior to that of the CoNiCrAlY substrate. The friction coefficient of NC25 was the lowest, and the width and depth of the worn surface were also the smallest, indicating the best wear resistance. However, the wear rate and friction coefficient of NC30 were higher than those of the substrate, suggesting that the content of hard-phase NiCr-Cr_3_C_2_ needs to be controlled to achieve optimal hardness and wear resistance.

Figure 6 shows the worn surface morphologies of CoNiCrAlY and NC5~NC30 cladding layer. Samples mainly exhibit adhesive wear, and more white wear debris can be observed in Figure 6b–e. Figure 6f shows that the amount of white wear debris significantly decreased for NC25. In Figure 6g, the size of the white wear debris continues to increase, which may indicate the aggregation and growth of abrasive particles during wear.

The XRD patterns of the CoNiCrAlY and NC5~NC30 cladding layers after wear are shown in Figure 7. Their phase compositions are similar, and the primary components detected include γ/γ′ and three types of chromium carbide (Cr_3_C_2_, Cr_7_C_3_, and Cr_23_C_6_), as well as oxides (NiO, Cr_2_O_3_, Al_2_O_3_, and (Ni,Co)Cr_2_O_4_). The main difference is that with the increase in NiCr-Cr_3_C_2_ content, the metal oxide content first decreased, then increased, and finally decreased, with a peak at NC25. Moreover, it can be demonstrated that more spinel phase (Ni,Co)Cr_2_O_4_ is formed on the surface of NC25. These oxides form a protective film on the surface of the cladding layer, effectively reducing external friction and wear. Moreover, Cr_2_O_3_ has a protective effect and provides some lubrication [21]. These factors can significantly improve the wear resistance of the cladding layer.

### 3.3. Molten Salt Corrosion Behavior at 750 °C

#### 3.3.1. Surface Phase and Morphology Analysis at 750 °C

A 50 h corrosion test was conducted utilizing a mixture of 75 wt.% Na_2_SO_4_ and 25 wt.% NaCl. The XRD patterns of the CoNiCrAlY and NC25 coatings with the best comprehensive mechanical properties under molten salt corrosion at 750 °C are shown in Figure 8. It can be observed that the γ/γ′ phase still exists in the CoNiCrAlY coating. However, the β phase and chromium carbide can no longer be detected because the β phase has been oxidized to Al_2_O_3_, and the chromium carbide has been oxidized to Cr_2_O_3_. The γ/γ′ phase on the corrosion surface decreased after adding 25% NiCr-Cr_3_C_2_, while Cr_2_O_3_, NiO, and the spinel phase (Ni,Co)Cr_2_O_4_ significantly increased. This indicates that the formation of the surface oxide film and a new spinel phase is promoted for the addition of NiCr-Cr_3_C_2_.

Figure 9 shows the surface morphology and elemental composition of the CoNiCrAlY cladding layer after molten salt corrosion at 750 °C. After 50 h of corrosion, parts of the oxide layer on the surface of the coating began to spall. The surface is mainly composed of NiO, CoO, a small amount of Al_2_O_3_, and some sulfates. The interior of the coating is filled with pitting corrosion holes, exhibiting a spongy appearance, which is mainly due to the corrosive effect of molten sulfates. Initially, Cr_2_O_3_ forms on the surface of the coating due to the combined action of oxygen and Na_2_SO_4_ and is subsequently dissolved by NaCl, as follows:4Cr + O_2_ + SO_4_^2−^ = 2Cr_2_O_3_ + S^2−^(2)
2Cr_2_O_3_ + 5O_2_ + 8Cl^−^ = 4(CrO_4_)^2−^ + 4Cl_2_↑(3)

Molten Na_2_SO_4_ decomposes into Na_2_O and SO_3_, and the NaCl in the molten salt is oxidized to produce chlorine gas, as follows:Na_2_SO_4_ = Na_2_O + SO_3_(4)
2SO_3_ = 2SO_2_ + O₂↑(5)
2Cl^−^ + O_2_ + SO_2_ = SO_4_^2−^ + Cl_2_↑(6)

The released chlorine gas and oxygen diffuse inward after the surface oxide film ruptures, reacting with the base metal to form the chlorides CrCl_2_, NiCl_2_, and AlCl_3_. The volatile chlorides diffuse outward through the pores and voids in the coating, and upon contact with oxygen, oxides like Cr_2_O_3_ are formed and loosely deposited on the surface [22]. Cr_2_O_3_ continue to be corroded by chloride ions in the molten salt, thus crack and spall gradually. In the initial stages of corrosion, Cr_2_O_3_ is consumed by Na_2_SO_4_, forming Cr_2_(SO_4_)_3_ and Na_2_O. In an alkaline environment, Cr_2_O_3_ reacts with Na_2_O to form Na_2_CrO_4_, accelerating the corrosion of the oxide layer. Additionally, the formed SO_3_ reacts with surface CoO to generate the low-melting-point eutectic compound Na_2_SO_4_-CoSO_4_, causing pitting corrosion and making the surface oxide film more porous and looser [23].

Figure 10 illustrates the surface morphology and elemental composition of the NC25 cladding layer after molten salt corrosion at 750 °C. Compared with the CoNiCrAlY cladding layer at the same temperature, there is a diffuse distribution of Ni and a small amount of Co oxides on the surface, as well as some aggregated and spherical Cr_2_O_3_ and (Ni,Co)Cr_2_O_4_. The formation of Cr_2_O_3_ undergoes decarburization and oxidation of carbides in a complex and specific process. Cr atoms have formed carbides, so they cannot independently diffuse to the coating surface to react with oxygen. Oxygen atoms diffuse from the surface oxide layer to the interface between the oxide and carbide phases, making the carbide phase gradually convert into the oxide phase.

Generally, Ni and Co elements are rapidly oxidized to NiO and CoO, which cover the slowly forming Cr_2_O_3_, and a more stable spinel phase (Ni,Co)Cr_2_O_4_ is created through the solid-phase reaction. The reaction processes are represented in Equations (7) and (8).
Cr_2_O_3_ + NiO = NiCr_2_O_4_(7)
Cr_2_O_3_ + CoO = CoCr_2_O_4_(8)

It has been previously shown that (Ni,Co)Cr_2_O_4_ can grow stably on the Cr_2_O_3_ substrate, which may be due to the good adhesion and stability of the oxide film formed during corrosion.

#### 3.3.2. Cross-Sectional Morphology Analysis at 750 °C

Figure 11 shows the cross-sectional morphology and elemental analysis results of the CoNiCrAlY cladding layer after molten salt corrosion at 750 °C. From the cross-sectional morphology, it can be seen that the spongy loose surface pitting depth is about 50 μm. This is mainly due to the chemical reaction between Na_2_SO_4_ and the surface of the cladding layer in the early stage of molten salt corrosion, forming low-density NiO and Cr_2_O_3_ oxide films that cannot provide good protection. As the reaction deepens and metal chlorides continue to evaporate, the corrosion reaction gradually penetrates into the cladding layer. At this point, the (CrO_4_)^2−^ formed, as in Equation (3), gradually diffuses to the surface of the cladding layer and reacts again with the oxidant in the molten salt, forming relatively loose Cr_2_O_3_. Oxygen channels are formed by this process on the surface, allowing the molten salt to penetrate the cladding layer more smoothly, thereby accelerating the corrosion of the cladding layer by molten salt. From the map scan results, it can be seen that during the molten salt corrosion, the elemental content inside the cladding layer remains relatively stable, proving that the interactions among elements in the CoNiCrAlY cladding layer can effectively resist molten salt corrosion. Meanwhile, the main protective oxide, Al_2_O_3_, effectively prevents the corrosion by molten salt.

The cross-sectional morphology and elemental analysis results for the NC25 cladding layer after molten salt corrosion at 750 °C are shown in Figure 12. From the cross-sectional morphology, it can be seen that the density of the surface oxide film increased, and no obvious extended pits are observed. This may be due to the sufficient oxidation reaction at high temperature, making the NiO and Cr_2_O_3_ oxide films more complete and denser. From the map scan results, a distinct layering phenomenon can be detected in the cladding layer, as follows: the outermost layer is mainly Cr_2_O_3_, the second layer is a mixed layer of NiO and Al_2_O_3_, and the rest is the matrix. This is due to the peeling of the surface Cr_2_O_3_ oxide film, and already a small amount of Al is consumed rapidly, forming an aluminum-poor zone. When the aluminum-poor zone increases to a certain thickness, the diffusion rate of Al cannot keep up with the consumption rate and diffuse to the surface, thus forming Al_2_O_3_ in the second layer.

Comparing the XRD patterns, surface microstructures, and elemental analysis results at 750 °C, it can be seen that after adding 25% NiCr-Cr_3_C_2_ the surface condition of the cladding layer after molten salt corrosion is improved, the number of pits is significantly reduced, and the corrosion resistance is enhanced.

#### 3.3.3. Molten Salt Corrosion Kinetics at 750 °C

The corrosion weight loss curves of the CoNiCrAlY and NC25 cladding layers at 750 °C are shown in Figure 13. In the range of 0~5 h, the cladding layer quickly lost weight, and then the weight loss rate gradually slowed down. During the corrosion process, the average corrosion weight loss rate of the CoNiCrAlY cladding layer was 1.558 mg/cm^2^, while the average corrosion weight loss rate of the NC25 cladding layer was 1.049 mg/cm^2^. This indicates that the molten salt corrosion resistance of the CoNiCrAlY cladding layer was significantly improved by the addition of 25 wt.% NiCr-Cr_3_C_2_.

The low-temperature hot corrosion mechanism at 750 °C can be summarized as follows: The melting point of the 75 wt.% Na_2_SO_4_ + 25 wt.% NaCl mixed salt is about 620 °C; thus, it is in a molten state during the experiment [24]. According to the acid-base dissolution mechanism, because of the different acidity and alkalinity of molten salt, protective oxides Cr_2_O_3_ and Al_2_O_3_ are dissolved in different ways [25]. For the sulfate system, the following reaction occurs at 750 °C:CoO + SO_3_ = CoSO_4_(9)

The SO_3_ generated, as per Equation (4), is consumed by CoO on the surface, forming a low-melting-point eutectic compound Na_2_SO_4_-CoSO_4_, whose melting point is only about 565 °C, existing as a liquid molten salt film at 750 °C [23]. In this molten salt film, SO_3_ migrates inward as (S_2_O_7_)^2−^, and Co^2+^, SO_4_^2−^, and dissolved SO_2_ migrate outward, causing pits and making the surface oxide film more porous and looser [26]. Moreover, beneficial elements are dissolved by these low-melting-point eutectics, and the formation or repair of Cr_2_O_3_ or Al_2_O_3_ oxide films is hindered, causing cobalt-based alloys to corrode rapidly. 

During the corrosion process, although no significant sulfides and pitting phenomena appear inside the NC25 cladding layer, a small number of internal oxidation pores form under the oxide film. The excellent corrosion resistance of the NC25 cladding layer is due to the rapid formation of Cr_2_O_3_ and (Ni,Co)Cr_2_O_4_, effectively preventing the formation of low-melting-point liquid Na_2_SO_4_-CoSO_4_ and forming a dense oxide film in the molten salt environment, improving the corrosion and oxidation resistance of the cladding layer. Furthermore, the protective film of Al_2_O_3_ is generated mainly through the Al elements in β-NiAl; thus, the content of β-NiAl and the consumption rate of the Al elements to some extent determine the corrosion resistance of the cladding layer. These factors together caused the cladding layer with 25 wt.% NiCr-Cr_3_C_2_ to exhibit good corrosion resistance in the molten salt environment at 750 °C.

### 3.4. Molten Salt Corrosion Behavior at 950 °C

#### 3.4.1. Surface Phase and Morphology Analysis at 950 °C

The XRD patterns of the CoNiCrAlY and NC25 cladding layers after molten salt corrosion at 950 °C are shown in Figure 14. It can be seen that the CoNiCrAlY cladding layer is severely corroded at 950 °C, with only γ/γ′ and NiO phases remaining, and the Cr_2_O_3_ phase is no longer detected. The phase composition of the NC25 cladding layer is similar to that after molten salt corrosion at 750 °C, indicating that the NC25 cladding layer has good corrosion resistance and can maintain a relatively stable phase composition at high temperatures. The matrix γ/γ′, two oxides NiO and Cr_2_O_3_, and the newly formed spinel phase (Ni,Co)Cr_2_O_4_ are detected, with higher intensity peaks of NiO and lower intensity peaks of Cr_2_O_3_.

Figure 15 shows the surface morphology and elemental composition of the CoNiCrAlY cladding layer after molten salt corrosion at 950 °C. It can be seen that at 950 °C, the surface of the cladding layer became more porous with more corrosion holes extending inward under the action of chloride ions. The outer surface oxide film is mainly composed of NiO with a small amount of Co and Al, and most of Cr_2_O_3_ has peeled off, indicating that the surface of the CoNiCrAlY cladding layer is heavily oxidized under high-temperature molten salt corrosion.

Figure 16 shows the surface morphology and elemental composition of the NC25 cladding layer after molten salt corrosion at 950 °C. Instead of obvious corrosion holes on the surface, a large number of spinel phases were generated at 950 °C; thus, the corrosion resistance of the NC25 cladding layer is further confirmed. The chemical composition of this spinel phase was detected to be mainly O, Cr, Co, and Ni, and through further composition analysis, it can be judged to be (Ni,Co)Cr_2_O_4_. The (Ni,Co)Cr_2_O_4_ phase has high thermal stability and good corrosion resistance, effectively resisting the erosion of corrosive substances in high-temperature environments. The (Ni,Co)Cr_2_O_4_ phase is partly distributed in spherical aggregates (Figure 16a) with some unformed spinel structures on the spheres, and mostly dispersed on the surface of the cladding layer (Figure 16b), indicating that the cladding layer has relatively uniform corrosion resistance.

Comparing the XRD patterns, surface microstructures, and elemental analysis results at 950 °C, it can be found that the CoNiCrAlY cladding layer is more likely to be corroded at high temperatures. In contrast, the NC25 cladding layer has a more stable phase composition and corrosion resistance at high temperatures, supporting its performance in practical applications.

#### 3.4.2. Cross-Sectional Morphology Analysis at 950 °C

Figure 17 shows the cross-sectional morphology and elemental analysis results of the CoNiCrAlY cladding layer after molten salt corrosion at 950 °C. Compared with its state at 750 °C, the cross-sectional pits are significantly smaller, indicating that the CoNiCrAlY has better molten salt corrosion resistance at higher temperatures because of reasonable elemental design. From the map scan results, it can be seen that the upper layer consists mainly of Ni, O, and Cr elements, which are from the NiO and Cr_2_O_3_ oxide films generated during corrosion that have not completely peeled off. A small amount of Al element is detected in the middle of the cladding layer in the form of Al_2_O_3_. The lower layer significantly increased in S content, in addition to Ni and Cr, and is the region heavily corroded by Na_2_SO_4_. The reason for the internal sulfurization may be that the loose Cr_2_O_3_ oxide film generated during corrosion continues to dissolve and fracture at high temperatures, and the film-forming elements Cr and Al are heavily consumed, resulting in the rapid diffusion of S along the pores into the interior of the substrate, forming sulfides such as CrS [27].

The cross-sectional morphology and elemental analysis results of the NC25 cladding layer after molten salt corrosion at 950 °C are shown in Figure 18. It can be seen that the upper layer is mainly composed of Al_2_O_3_ and Cr_2_O_3_, and the internal γ/γ′ phase is well protected. Compared with the CoNiCrAlY, the S content inside the NC25 decreases, and internal sulfurization is reduced. This may be attributed to the fact that Na_2_SO_4_ is already heavily consumed when dissolving surface Cr_2_O_3_, forming a large amount of spinel phase, the corrosion effect of Na_2_SO_4_ is reduced and the inward diffusion of S is relieved.

#### 3.4.3. Molten Salt Corrosion Kinetics at 950 °C

The corrosion weight loss curves of the CoNiCrAlY and NC25 cladding layers at 950 °C are shown in Figure 19. It can be seen that the weight loss rate was relatively slow in the initial corrosion period of 0~20 h, Cr and Al are slowly oxidized on the surface, and the oxide film gradually generated. The stress was raised and some local cracks were produced, but there was no loss in the overall mechanical integrity of the sample. Meanwhile, the acidity and alkalinity of Na_2_SO_4_ changed with the molten salt environment. In the mid-corrosion period of 20~35 h, the surface oxide film formed in large amounts and then dissolved by Na_2_SO_4_, and the film-forming elements Cr and Al below the oxide film–matrix interface were heavily consumed. These lead to penetration of S into the cladding layer, where the weight loss rate is significantly accelerated and corrosion is deepened. In the late corrosion period of 35~50 h, the surface oxide film peels off on a large scale, S continues to penetrate downward to corrode the substrate, and sulfides form in the interior. The sample was severely corroded, and the overall mechanical integrity was damaged. The weight loss of the CoNiCrAlY and NC25 cladding layers showed a similar trend as the corrosion proceeded. However, the weight loss was more severe in the CoNiCrAlY, with an average corrosion weight loss rate of 9.428 mg/cm^2^, while the NC25 was corroded relatively lightly at a rate of 3.242 mg/cm^2^. The CoNiCrAlY was more easily damaged during corrosion, while the NC25 was relatively stable, indicating that the molten salt corrosion resistance of the cladding layer at 950 °C can be significantly improved by the addition of 25 wt.% NiCr-Cr_3_C_2_.

Moreover, compared with the corrosion results at 750 °C, the molten salt corrosion resistance of the substrate significantly improved by 25 wt.% NiCr-Cr_3_C_2_, indicating that the cladding layer with this composition can perform better at 950 °C. These data and information will serve as an important reference for research and in the application of corrosion-resistant cladding layers.

By comparing the molten salt corrosion results of CoNiCrAlY and NC25 cladding layers at 950 °C, it can be seen that in addition to oxide films, the substrate can be effectively protected by the (Ni,Co)Cr_2_O_4_ phase formed during corrosion [28]. As shown in Figure 20, taking NiCr_2_O_4_ as an example, it is a typical Fd3m spinel structure, with the crystal cell composed of tetrahedral clusters [NiO_4_] and octahedral clusters [CrO_6_]. The (Ni,Co)Cr_2_O_4_ phase is structurally stable at high temperature, with the formation enthalpy ∆H at 950 °C of NiCr_2_O_4_ being −1212.23 kJ/mol and that of CoCr_2_O_4_ being −1263.16 kJ/mol [29].

The high-temperature hot corrosion mechanism at 950 °C can be summarized as follows: because of the surface tension and capillary effects, after the molten salt penetrates the porous oxide layer, smaller pores are filled first, leading to very fast corrosion speed, with almost no continuous oxide layer between the oxide layer and the substrate. At high temperatures, the diffusion speed of gases and ions is faster. Molten Na_2_SO_4_ is dissolved in alkaline solution, according to Equation (4), at 950 °C, and Cr is first oxidized by O_2_ produced by molten salt decomposition to Cr_2_O_3_. As the equilibrium, as shown by Equation (2), shifts to the right, the partial pressure of S in the oxide film increases, with S_2_O_7_^2−^ participating in the reaction at the interface between the oxide film and the molten salt, as the main oxidant, as follows:(S_2_O_7_)^2−^ = SO_4_^2−^ + SO_3_(10)
SO_4_^2−^ + 2e^−^ = 4O_2_^−^ + S^2−^(11)

S^2−^ penetrates the cladding layer, reacting with metals or metal oxides to form sulfides. As the reaction progresses, the alkaline Na_2_O in the molten salt continue to increases, and Cr_2_O_3_ dissolves into the molten salt [30]. The reaction process is represented by Equation (12), as follows:Cr_2_O_3_ + O^2−^ = (Cr_2_O_4_)^2−^(12)

It is indicated in the Rapp–Goto molten salt model that if there is a negative gradient in the dissolution of protective metal oxides in the molten salt, hot corrosion may continue [31]. This means that certain metal oxides gradually dissolve and precipitate in the molten salt, resulting in the metal substrate is exposed to and corroded by the molten salt. The equilibrium of Equation (12) shifts to the right attributed to the high content of Cr_2_O_3_ in the NC25 cladding layer, more (Cr_2_O_4_)^2−^ is generated, and then the spinel phase (Ni,Co)Cr_2_O_4_. This process causes the alkalinity of the molten salt to gradually exceed the initial value of corrosion and stabilize at a certain level. During this process, the negative gradient in the dissolution of protective metal oxides gradually disappears and reaches saturation. Oxygen in the environment can reach the interface between the substrate and the oxide film as the old oxide film peels off; new dense Al_2_O_3_ and Cr_2_O_3_ films are generated, and further corrosion is prevented. An illustration of the corrosion process is shown in Figure 21.

It is shown in the above tests that distinct layering can be observed in the corroded cladding layer. The upper layer is NiO and some loose Cr_2_O_3_ layers; the middle layer is mostly Al_2_O_3_, Cr_2_O_3_, and spinel phase (Ni,Co)Cr_2_O_4_; and the lower layer is the substrate and sulfides. Cr and Al play crucial roles in the corrosion resistance of the cladding layer, closely related to their characteristic of readily reacting with oxygen to form oxides. Because of the strong affinity of Cr and Al with oxygen in the initial stage of oxidation, they rapidly combine with oxygen to form oxides, leading to a significant increase in the weight of the cladding layer. However, as the oxidation process continues, further oxygen penetration is effectively prevented by these oxides. Some of the Cr elements are converted into volatile Cr_2_O_3_ during oxidation, which slows down the oxidative weight gain of the cladding layer and enables the cladding layer to remain stable for a longer period of time. The more stable Al_2_O_3_ film plays a major role in the cladding layer, and its relatively slow growth kinetics make it difficult for oxygen to diffuse through it. Therefore, the Al_2_O_3_ film can maintain substantial inertness even in high-temperature and oxidative atmospheres. This excellent oxidation resistance allows the NC25 cladding layer to maintain a stable performance even in harsh environments, and its service life is extended.

## 4. Conclusions

The effects of the NiCr-Cr_3_C_2_ content on the microstructures, mechanical properties, and high temperature corrosion resistance of CoNiCrAlY cladding layers fabricated by microbeam plasma technology were investigated, and its corrosion mechanism was clarified in this paper. The main conclusions are as follows:With the addition of NiCr-Cr_3_C_2_, the layer grain size firstly increases and then decreases, and the dispersion and granular width of the β phase are promoted.The NC25 cladding layer had the highest average hardness at 348.2 HV_0.3_, as well as the lowest average friction coefficient and wear rate at 0.4751 and 0.4528 × 10^−6^ mm^3^/N·m, respectively. The wear mechanism is the surface of the cladding layer undergoing oxidative wear during friction with the formation of a dense protective Al_2_O_3_ film, together with Cr_2_O_3_ and (Ni,Co)Cr_2_O_4_. Additionally, Cr_2_O_3_ has certain lubricative function, which improves the layer’s tribological performance.Under molten salt corrosion at 750 °C, the cladding layer exhibited typical low-temperature hot corrosion characteristics, with pitting holes extending inward on the surface due to the rapid corrosion by Na_2_SO_4_. The rapid formation of Cr_2_O_3_ and (Ni,Co)Cr_2_O_4_ in the NC25 cladding layer effectively inhibited the formation of low-melting-point eutectic Na_2_SO_4_-CoSO_4_, which reduced the average weight loss rate by 32.7%.Under molten salt corrosion at 950 °C, the cladding layer exhibited typical high-temperature hot corrosion characteristics, with local Al depletion and sulfide formation. The high content of Cr_2_O_3_ in the NC25 cladding layer promoted the formation of the spinel phase (Ni,Co)Cr_2_O_4_, inhibiting the corrosion effect of Na_2_SO_4_ and ion diffusion, significantly improving the stability of the oxide film, which reduced the average weight loss rate by 65.6%. Moreover, at 950 °C, the improvement in the corrosion resistance of the NC25 cladding layer compared with the matrix was far greater than at 750 °C, indicating it is more suitable for service at 950 °C.

## Figures and Tables

**Figure 1 materials-17-04249-f001:**
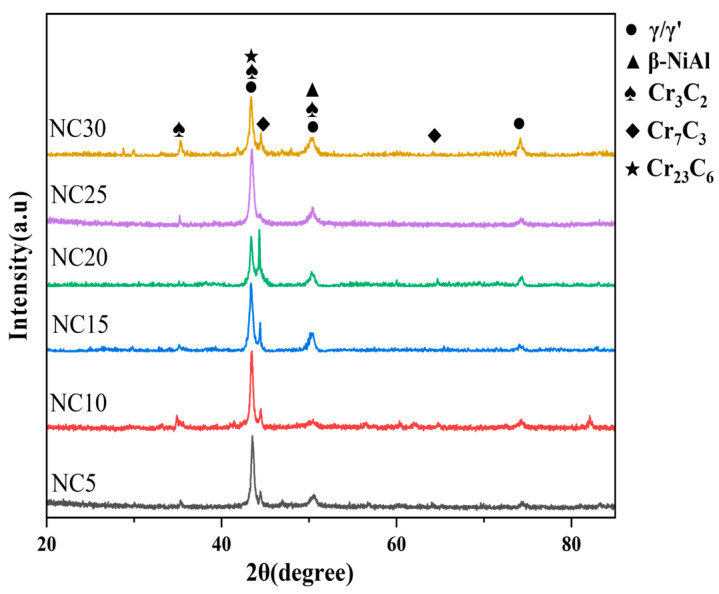
XRD patterns of NC5~NC30.

**Figure 2 materials-17-04249-f002:**
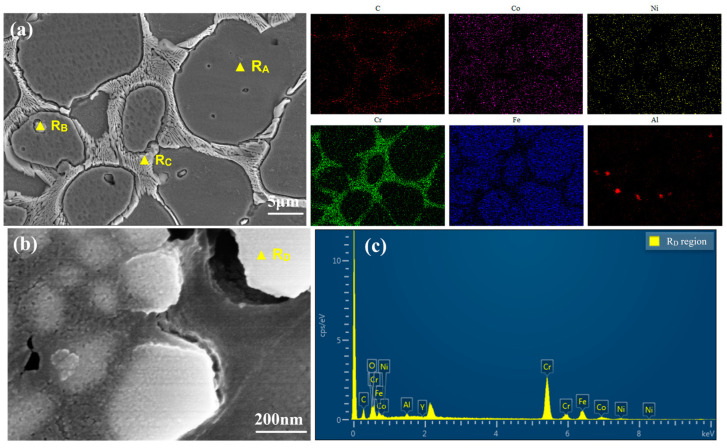
(**a**) Mapping scan results of NC25; (**b**) microstructural morphologies of the intergranular regions; (**c**) EDS spectrum of R_D_.

**Figure 3 materials-17-04249-f003:**
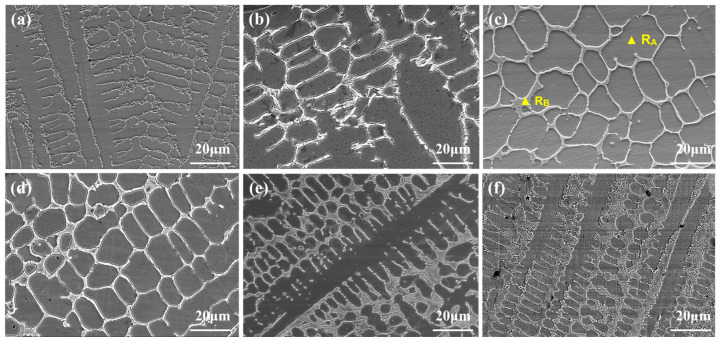
Microstructural morphologies of the cladding layers: (**a**) NC5; (**b**) NC10; (**c**) NC15; (**d**) NC20; (**e**) NC25; (**f**) NC30.

**Figure 4 materials-17-04249-f004:**
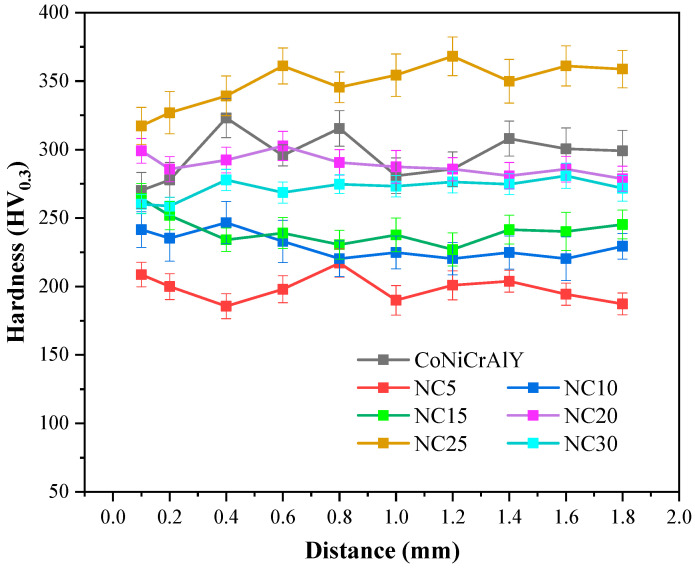
The micro-hardness curves of the CoNiCrAlY and NC5~NC30 cladding layer sections.

**Figure 5 materials-17-04249-f005:**
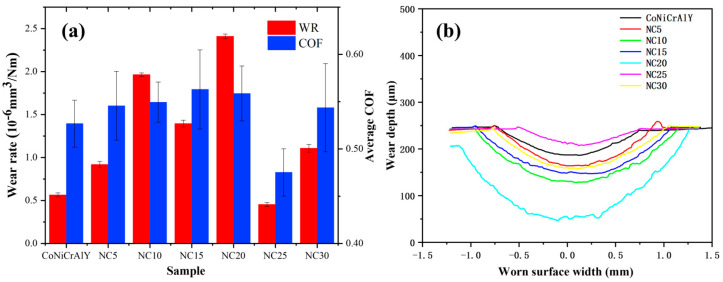
(**a**) Average COF and wear rate; (**b**) 2D images of worn surface morphology.

**Figure 6 materials-17-04249-f006:**
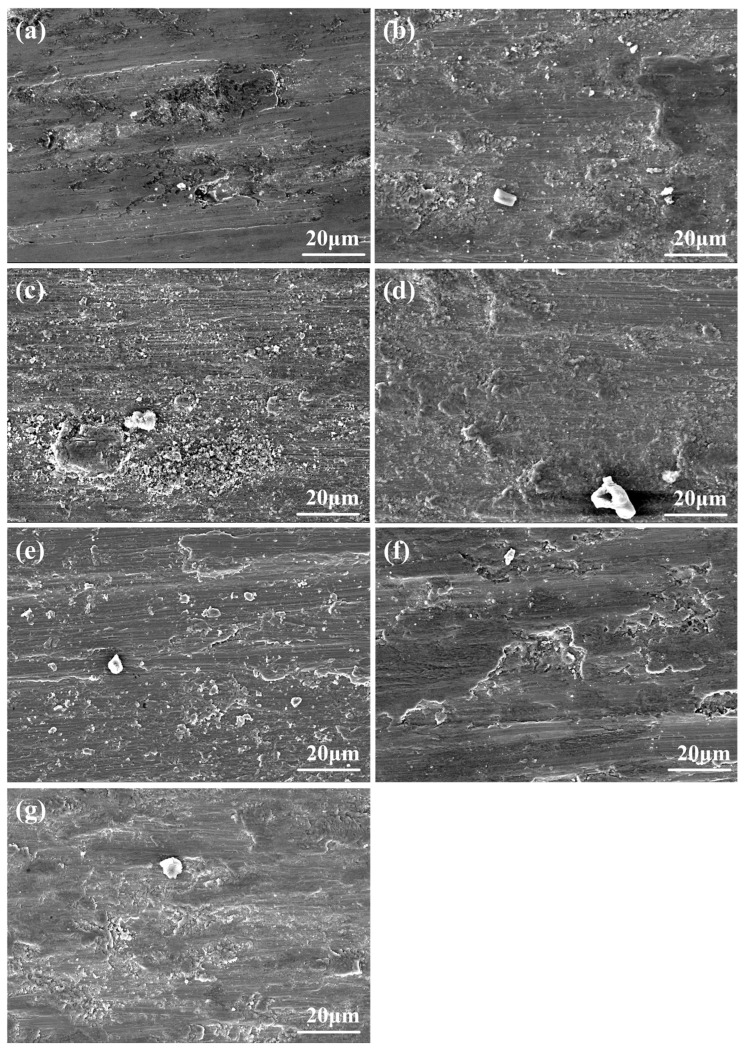
Worn surface morphologies: (**a**) CoNiCrAlY; (**b**) NC5; (**c**) NC10; (**d**) NC15; (**e**) NC20; (**f**) NC25; (**g**) NC30.

**Figure 7 materials-17-04249-f007:**
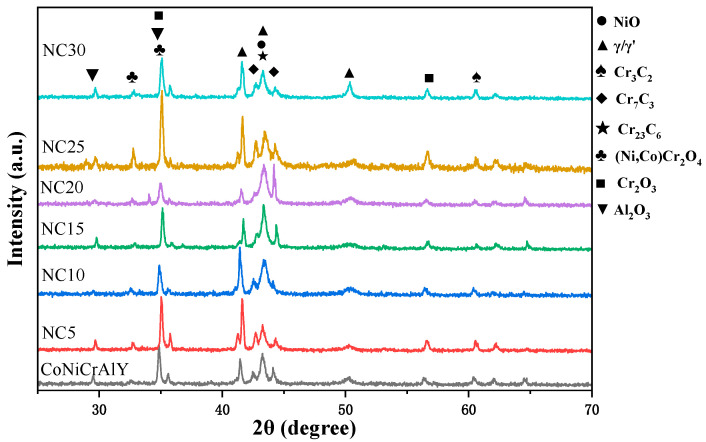
XRD patterns of CoNiCrAlY and NC5~NC30 after wear.

**Figure 8 materials-17-04249-f008:**
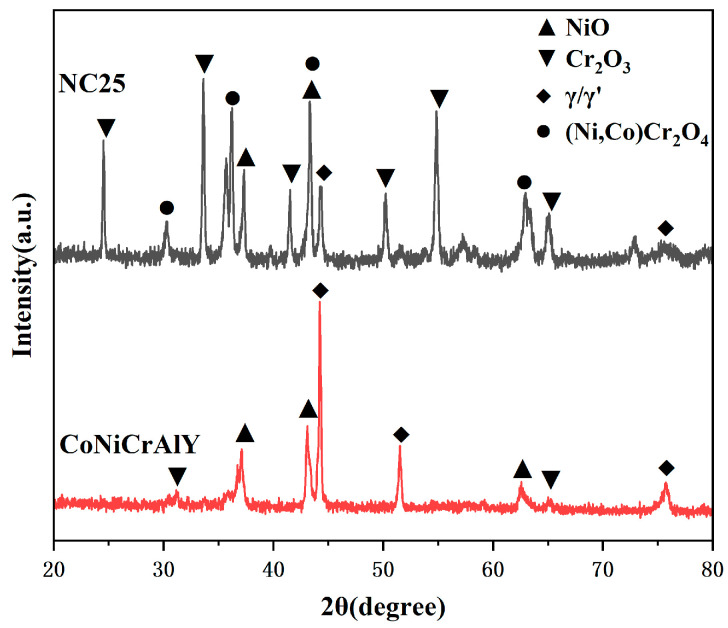
The XRD pattern of molten salt corrosion products at 750 °C.

**Figure 9 materials-17-04249-f009:**
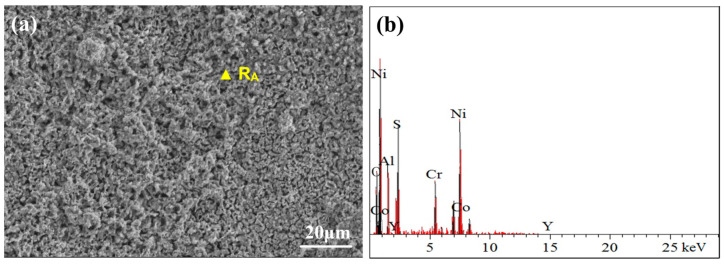
(**a**) The surface morphology of the CoNiCrAlY cladding layer after molten salt corrosion at 750 °C; (**b**) EDS spectrum of R_A_.

**Figure 10 materials-17-04249-f010:**
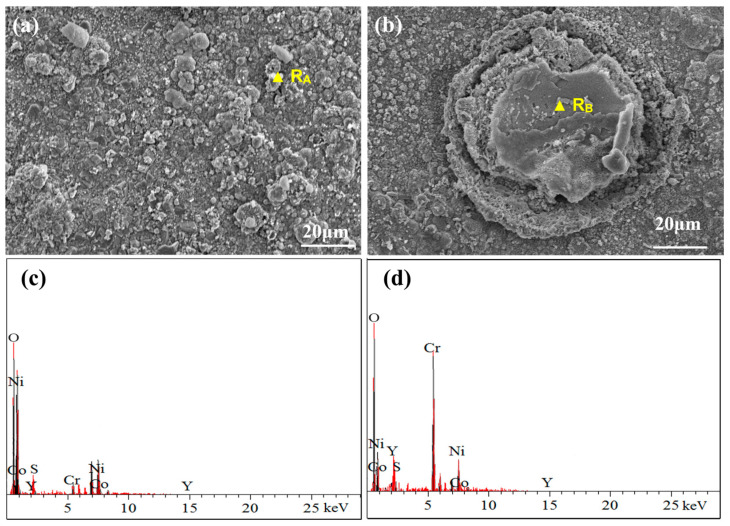
(**a**,**b**) Surface morphology of the NC25 cladding layer after molten salt corrosion at 750 °C; (**c**) EDS spectrum of R_A_; (**d**) EDS spectrum of R_B_.

**Figure 11 materials-17-04249-f011:**
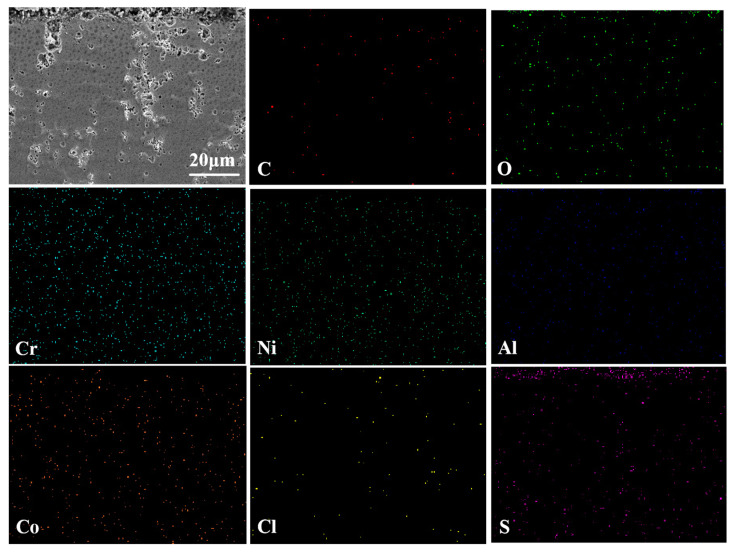
The cross-sectional morphology of the CoNiCrAlY cladding layer after molten salt corrosion at 750 °C and the EDS map scan results.

**Figure 12 materials-17-04249-f012:**
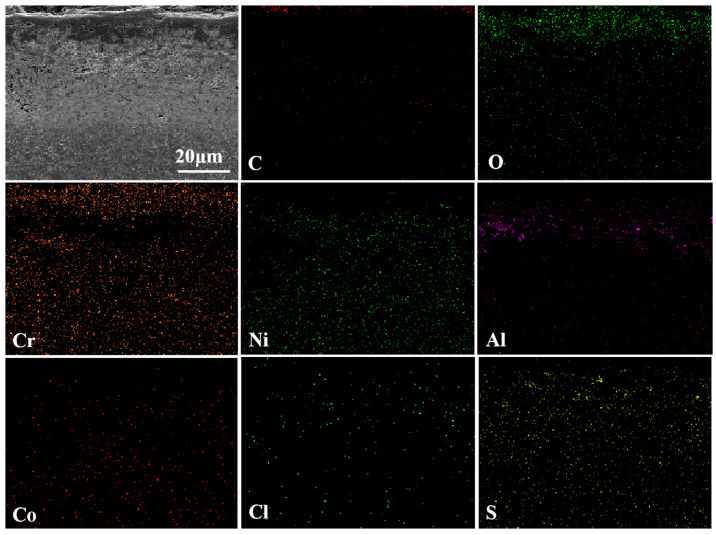
The cross-sectional morphology of the NC25 cladding layer after molten salt corrosion at 750 °C and EDS map scan results.

**Figure 13 materials-17-04249-f013:**
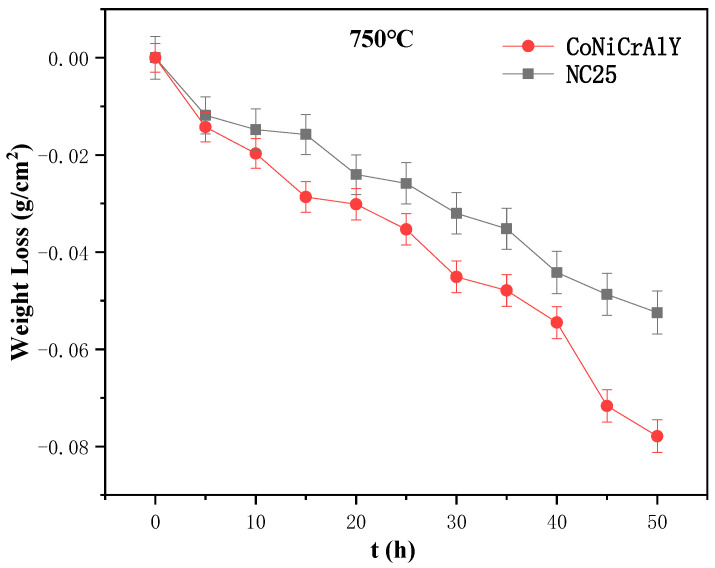
Corrosion weight loss curves of CoNiCrAlY and NC25 at 750 °C.

**Figure 14 materials-17-04249-f014:**
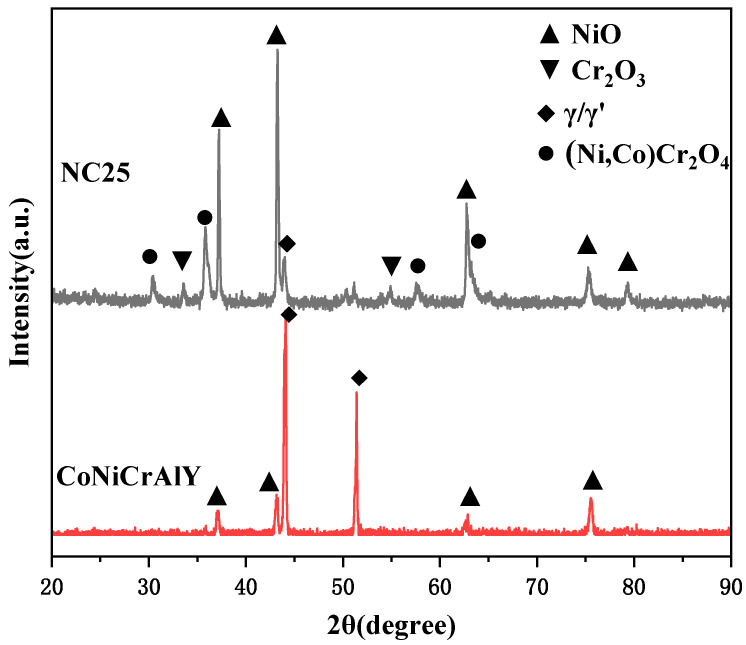
The XRD patterns of molten salt corrosion products at 950 °C.

**Figure 15 materials-17-04249-f015:**
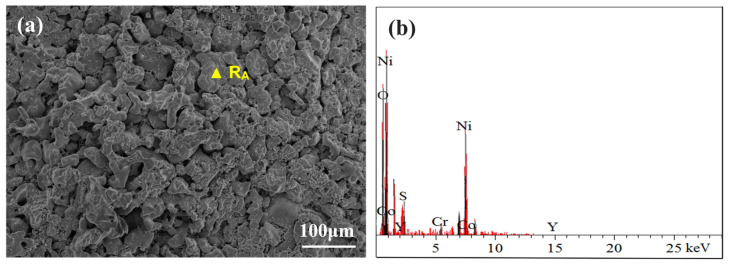
(**a**) The surface morphology of the CoNiCrAlY cladding layer after molten salt corrosion at 950 °C; (**b**) EDS spectrum of R_A_.

**Figure 16 materials-17-04249-f016:**
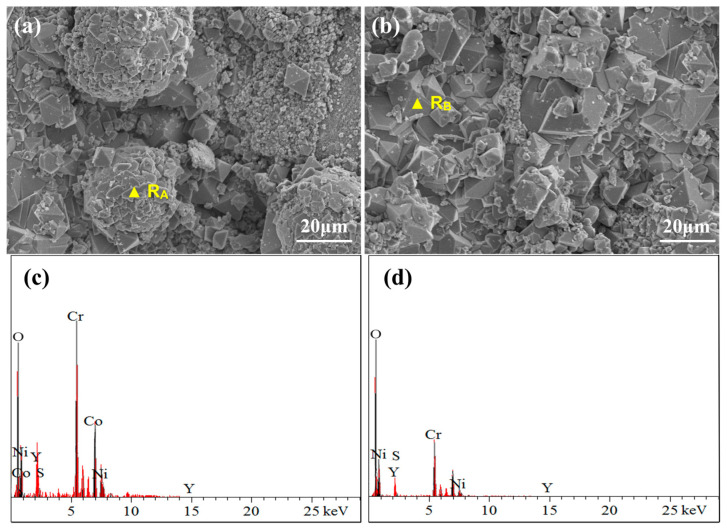
(**a**,**b**) Surface morphology of the NC25 cladding layer after molten salt corrosion at 950 °C; (**c**) EDS spectrum of R_A_; (**d**) EDS spectrum of R_B_.

**Figure 17 materials-17-04249-f017:**
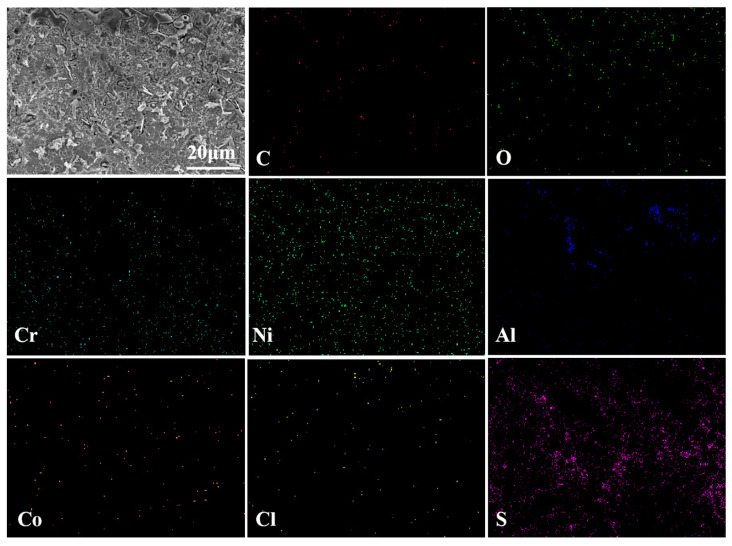
The cross-sectional morphology of the CoNiCrAlY cladding layer after molten salt corrosion at 950 °C and EDS map scan results.

**Figure 18 materials-17-04249-f018:**
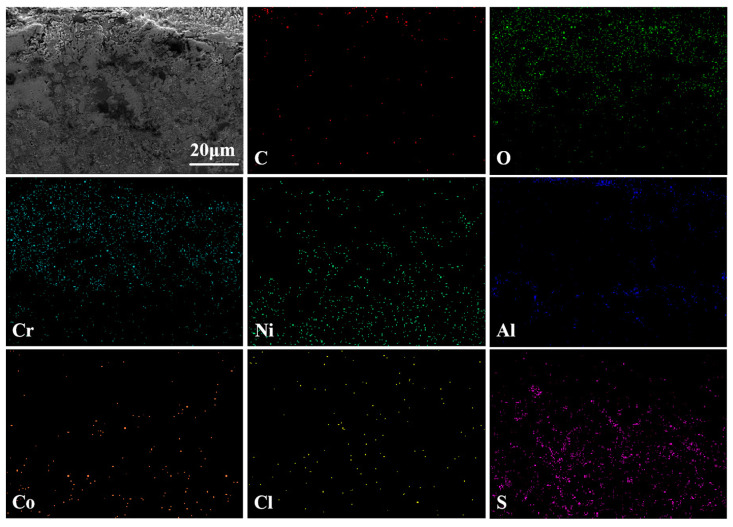
The cross-sectional morphology of the NC25 cladding layer after molten salt corrosion at 950 °C and EDS map scan results.

**Figure 19 materials-17-04249-f019:**
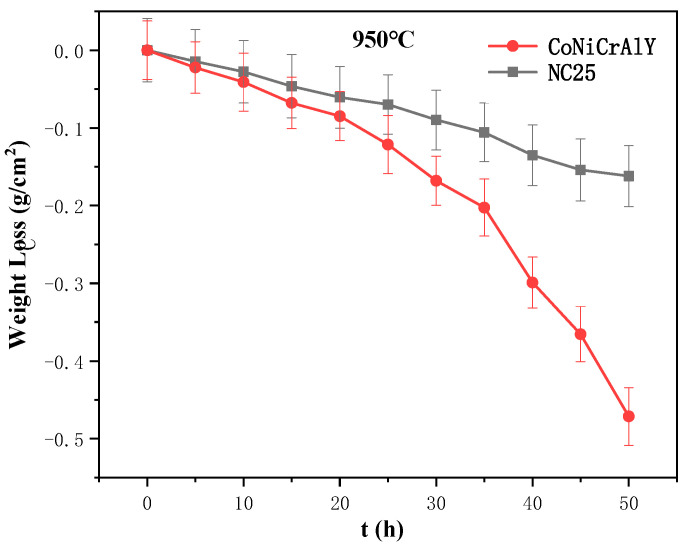
Corrosion weight loss curves of CoNiCrAlY and NC25 at 950 °C.

**Figure 20 materials-17-04249-f020:**
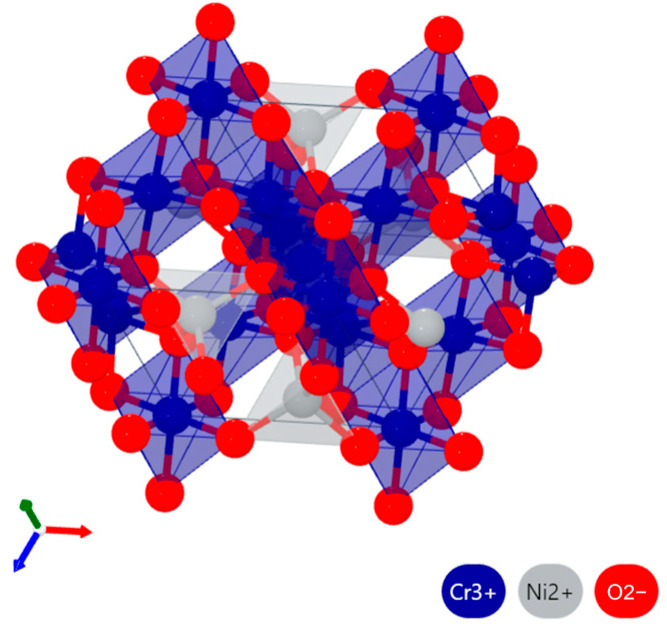
Schematic diagram of the NiCr_2_O_4_ crystal structure.

**Figure 21 materials-17-04249-f021:**
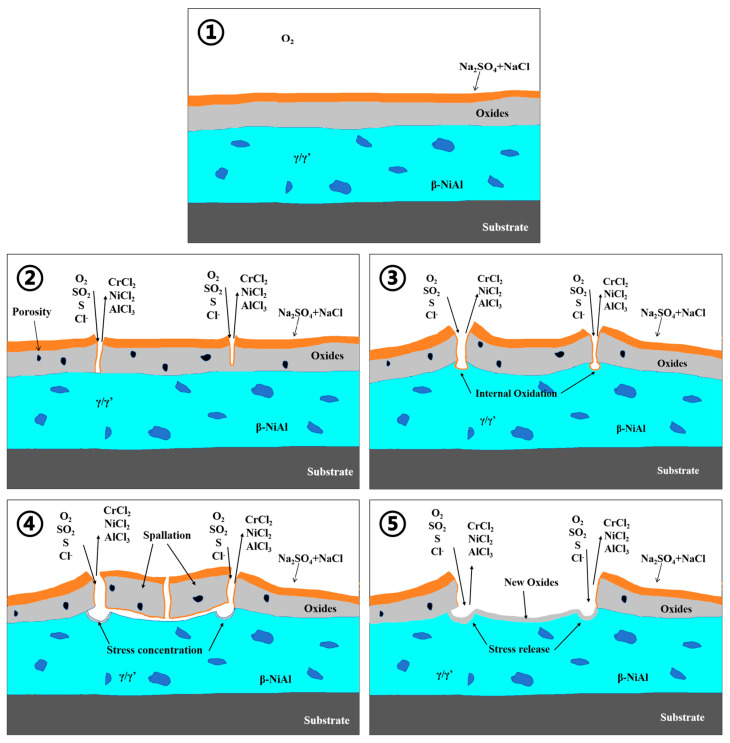
Illustration of the molten salt corrosion process.

**Table 1 materials-17-04249-t001:** Composition of CoNiCrAlY (wt.%).

Element	Co	Ni	Cr	Al	Y
Composition	38.5	32	21	8	0.5

**Table 2 materials-17-04249-t002:** Compositions of cladding layers (wt.%).

No.	CoNiCrAlY	NiCr-Cr_3_C_2_
CoNiCrAlY	100	0
NC5	95	5
NC10	90	10
NC15	85	15
NC20	80	20
NC25	75	25
NC30	70	30

**Table 3 materials-17-04249-t003:** Cladding process parameters.

Current I/A	Voltage U/V	Welding Velocity v/(mm·s^−1^)	Plasma Gas Pressure P/(MPa)	Shielding Gas Flow F/(L·min^−1^)	Plasma Arc Height (mm)	Nozzle Diameter (mm)
22.5	20	0.6	0.4	8	4	2.4

**Table 4 materials-17-04249-t004:** Tribological test parameters.

Temperature (°C)	Applied Load (N)	Frequency (Hz)	Stroke Length (mm)	Testing Time (min)
25	20	5	10	30

**Table 5 materials-17-04249-t005:** Proportions of the phase contents of the NC5~NC30 cladding layers (wt.%).

	γ/γ′	β-NiAl	Cr_3_C_2_	Cr_7_C_3_	Cr_23_C_6_
NC5	54.6	1.4	17.9	7.4	18.7
NC10	46.3	7.3	19.3	9.5	17.6
NC15	46.6	7.6	16.5	14.7	14.6
NC20	38.4	12.2	13.5	26.2	9.6
NC25	41.7	15.8	24.6	10.2	7.7
NC30	49.6	6.7	16.9	16.4	10.3

## Data Availability

The original contributions presented in the study are included in the article, further inquiries can be directed to the corresponding author.

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
