# Peer review of "Fabrication of Co-Based Cladding Layer by Microbeam Plasma and Its Corrosion Mechanism to Molten Salt"

_materials, 2024, doi:10.3390/ma17174249_

Round 1

Reviewer 1 Report

Comments and Suggestions for Authors

The manuscripts investigates the effect of NiCr-Cr3C2 content in CoNiCrAlY cladding layers, obtained by microbeam plasma on Inconel 625 substrate, on their microstructure, mechanical properties and high temperature corrosion resistance in molten salts.

The manuscript idea is novel and very important for future molten salts applications. The experimental design is very well planned, the results are broadly presented and discussed. Overall, the manuscript quality is very high.

I may not offer any recommendation for further improvement of the manuscript, and so I recommend its publication in the present form.

Reviewer 2 Report

Comments and Suggestions for Authors

I find the manuscript very novel in its approach to addressing issues of thermal corrosion. The design and development of high-temperature corrosion-resistant coatings is well articulated. The study has been fully investigated and the results are well-reported. However, a few points must be addressed before the manuscript is accepted. The introduction needs to be revised in such a way that the motive of the study becomes unambiguous.  Some of the minor errors noted are as follows.

The aims and objectives of this study need to be revised. There is a need to elaborate on what good properties mean in line 53.

The aim of the study as written in lines 52-53 is not clearly stated as shown in the abstract.

The authors mentioned the reduction of maintenance costs in line 57 as one of the objectives but there is no proof to support this idea.

Line 51, Reference 18 needs to be confirmed.

Line 48-50, the authors should provide the temperature range for the thermal corrosion which was only mentioned to be in two categories i.e. low and high temperature. 

Comments on the Quality of English Language

English language is fine only needs minor revisions.

Reviewer 3 Report

Comments and Suggestions for Authors

The authors have presented the paper entitled "Fabrication of Co-based cladding layer by micro-beam plasma and its corrosion mechanism to molten salt". I have to say that the paper was very interesting to read. It is well presented and organized and all together it gives a great value to the field.

However some minor aspects can be improved.

- As a personal opinion when a paragraph starts with the word "Figure" it has to be the complete word and not only the abbreviation "Fig. X". The authors in the paper have several paragraphs along the paper where they use the "Fig.X" abbreviation as beginning of the sentence and I believe, this is not correct.

- In the Figure caption "Figure 10. The surface morphology of the NC25 cladding layer after molten salt corrosion at 750 °C. (a) Magnify 1000 times; (b) Magnify 500 times." I believe it is not correct and neither the scale bar on the figures nor the magnification. On that last 2 points I'm not 100% sure, but is just an educated hunch. However, the figure 10 corresponds to 4 micrographs and therefore it must be 4 different statements or at leas to mention the 4 labels (a, b, c and d).

-And one last point, the authors have mentioned the SEM equipment used but nothing about the XRD equipment. Please do so.

Comments on the Quality of English Language

In general good , just a few moments.

The initial Fig. to be changed to Figure at the beginning of the paragraphs and some minor mistakes.

Reviewer 4 Report

Comments and Suggestions for Authors

The article presents the results of the study of the microstructure of CoNiCrAlY coatings and their wear resistance and corrosion resistance in molten salts.

The article is well organized, but requires explanations and improvement of the analysis and interpretation of the research results.

Below I present my comments on the article.

The introduction describes the current state of knowledge with reference to References, but I believe that the authors should present more clearly the existing problem, which is important to solve and undertake this research.

The materials, procedures for producing test samples and the research methods have been sufficiently described.

Results and Discussion

Lines 114-122: EDS mapping presented in Figure 1 is not able to confirm the phases occurring in the structure, but only the presence of given elements in micro-areas. Therefore, this discussion should be correlated with the results of XRD analysis or supported by reference to References.

Lines 123-129: The authors indicate the presence of various chromium carbides. The presented EDS analysis results in Table 5 are not able to effectively confirm this. In addition, in Table 5 the authors present the quantitative carbon content. The EDS method is not suitable for determining the quantitative carbon content. The percentage C content cannot be reliable here. The EDS method only allows for the qualitative determination of the presence of carbon. Therefore, I believe that it is better to include the EDS spectrum than to provide unreliable quantitative results of the elemental content.

Figure 2: The structural components indicated in the text should be marked on the SEM images, and the discussion regarding this figure should be correlated with the results of the EDS and XRD analysis.

The title of chapter 3.2 is completely incomprehensible. The title should characterize the content of the chapter. This title requires a complete change.

Figure 4: The microhardness values ​​should be specified as average values ​​from several measurements. Error bars are also required on the graph.

Figure 5a, Table 7: The average values ​​should have the measurement error indicated.

Table 8: The EDS method is not suitable for determining the quantitative carbon content, but also oxygen (see note earlier).

Chapter 3.3.1: The described corrosion mechanism requires a strong reference to References because the presented research results are not able to confirm this, e.g. on what basis do the authors conclude that oxygen atoms diffuse from the oxide surface, that the eutectic compound Na2SO4-CoSO4 is formed? On what basis do the authors formulate the course of the reaction? The SEM images should also indicate the structural components or places of their occurrence, which the authors list and describe their transformation in the text. I also maintain my comment regarding the determination of the quantitative content of C and O (Tables 9 and 10). These comments also apply to chapter 3.3.2.

Figure 13: On what number of samples was this study carried out? The mass losses should be an average value and have a specified measurement error.

Chapter 3.4 - See comments to chapter 3.3.

Round 2

Reviewer 4 Report

Comments and Suggestions for Authors

The authors provided a revised article and responded to all my comments.

The authors improved the article in accordance with my comments and provided very substantive explanations.

However, I am still concerned about the interpretation of the EDS analysis. The EDS method is not suitable for the quantitative determination of oxygen and carbon, which is also pointed out by the authors in their answer. Therefore, I propose to refrain from presenting the quantitative chemical composition, which is included in the tables. It is better to include only the EDS light, which will allow for qualitative identification of the presence of elements. Moreover, I propose to completely abandon the interpretation in which the phase containing carbon is identified based on the atomic ratio of the elements.
